# STABLE WEIGHT DECAY REGULARIZATION

## ABSTRACT

Weight decay is a popular regularization technique for training of deep networks. Modern deep learning libraries mainly use $L_2$ regularization as the default implementation of weight decay. Loshchilov & Hutter (2018) demonstrated that $L_2$ regularization is not identical to weight decay for adaptive gradient methods, such as Adaptive Momentum Estimation (Adam), and proposed Adam with Decoupled Weight Decay (AdamW). However, we found that the popular implementations of weight decay, including $L_2$ regularization and decoupled weight decay, in modern deep learning libraries usually damage performance. First, the $L_2$ regularization is unstable weight decay for all optimizers that use Momentum, such as stochastic gradient descent (SGD). Second, decoupled weight decay is highly unstable for all adaptive gradient methods. We further propose the *Stable Weight Decay* (SWD) method to fix the unstable weight decay problem from a dynamical perspective. The proposed SWD method makes significant improvements over $L_2$ regularization and decoupled weight decay in our experiments. Simply fixing weight decay in Adam by SWD, with no extra hyperparameter, can outperform complex Adam variants, which have more hyperparameters.

## 1 INTRODUCTION

Weight decay is a popular and even necessary regularization technique for training deep neural networks that generalize well (Krogh & Hertz, 1992). People commonly use $L_2$ regularization as "weight decay" for training of deep neural networks and interpret it as a Gaussian prior over the model weights. This is true for vanilla SGD. However, Loshchilov & Hutter (2018) revealed that, when the learning rate is adaptive, the commonly used $L_2$ regularization is not identical to the vanilla weight decay proposed by Hanson & Pratt (1989):

$$\theta_t = (1 - \lambda_0)\theta_{t-1} - \eta g_t, \tag{1}$$

where $\lambda_0$ is the weight decay hyperparameter, $\theta_t$ is the model parameters at $t$-th step , $\eta$ is the learning rate, and $g_t$ is the gradient of the minibatch loss function $L(\theta)$ at $\theta_{t-1}$. Zhang et al. (2018) revealed three different roles of weight decay. However, the quantitative measures for weight decay are still missing.

Adaptive gradient methods that use Adaptive Learning Rate, such as AdaGrad (Duchi et al., 2011), RMSprop (Hinton et al., 2012), Adadelta (Zeiler, 2012) and Adam (Kingma & Ba, 2015), are a class of most popular methods to accelerate training of deep neural networks. Loshchilov & Hutter (2018) reported that, in adaptive gradient methods, the correct implementation of weight decay should be applied to the weights directly and decoupled from the gradients. We show the different implementations of decoupled weight decay and $L_2$ regularization for Adam in Algorithm 3.

It has been widely observed that adaptive gradient methods usually do not generalize as well as SGD (Wilson et al., 2017). A few Adam variants tried to fix the hidden problems in adaptive gradient methods, including AdamW Loshchilov & Hutter (2018), AMSGrad (Reddi et al., 2019) and Yogi (Zaheer et al., 2018). A recent line of research, such as AdaBound (Luo et al., 2019), Padam (Chen & Gu, 2018), and RAdam (Liu et al., 2019), believes controlling the adaptivity of learning rates may improve generalization. This line of research usually introduces extra hyperparameters to control the adaptivity, which requires more efforts in tuning hyperparameters.

Although popular optimizers have achieved great empirical success for training of deep neural networks, we discover that nonoptimal weight decay implementations have been widely used in

Table 1: Test performance comparison of optimizers. We report the mean and the standard deviations (as the subscripts) of the optimal test errors computed over three runs of each experiment. AdamS generalizes better than popular adaptive gradient methods significantly and often compares favorably with the baseline optimizer SGD.

| DATASET | MODEL | SGD | ADAMS | ADAM | AMSGRAD | ADAMW | ADABOUND | PADAM | YOGI | RADAM |
|---------|-------|-----|-------|------|---------|-------|----------|-------|------|-------|
| CIFAR-10 | RESNET18 | $5.01_{0.03}$ | $\textbf{4.91}_{0.04}$ | $6.96_{0.02}$ | $6.16_{0.18}$ | $5.08_{0.07}$ | $5.65_{0.08}$ | $5.12_{0.04}$ | $5.87_{0.12}$ | $6.01_{0.10}$ |
| | VGG16 | $6.42_{0.02}$ | $\textbf{6.09}_{0.11}$ | $7.31_{0.25}$ | $7.14_{0.14}$ | $6.59_{0.13}$ | $6.76_{0.12}$ | $6.15_{0.06}$ | $6.90_{0.22}$ | $6.56_{0.04}$ |
| CIFAR-100 | DENSENET121 | $\textbf{19.81}_{0.33}$ | $20.52_{0.26}$ | $25.11_{0.15}$ | $24.43_{0.09}$ | $21.55_{0.14}$ | $22.69_{0.15}$ | $21.10_{0.23}$ | $22.15_{0.36}$ | $22.27_{0.22}$ |
| | GOOGLENET | $21.21_{0.29}$ | $\textbf{21.05}_{0.18}$ | $26.12_{0.33}$ | $25.53_{0.17}$ | $21.29_{0.17}$ | $23.18_{0.31}$ | $21.82_{0.17}$ | $24.24_{0.16}$ | $22.23_{0.15}$ |

modern deep learning libraries, including PyTorch (Paszke et al., 2019), TensorFlow (Abadi et al., 2016), MXNet (Chen et al., 2015), and Chainer (Tokui et al., 2019). The nonoptimal implementations can lead to the unstable weight decay problem during training, which may harm performance seriously. We try to answer when the unstable weight decay problem happens and how to mitigate it. While Loshchilov & Hutter (2018) discovered that weight decay should be decoupled from the gradients, we further discovered that weight decay should be coupled with the effective learning rate. We organize our main findings as follows.

**1. The effect of weight decay can be interpreted as iteration-wisely rescaling the loss landscape and the learning rate at the same time.** We formulate the weight decay rate based on the rescaling ratio per unit stepsize. We call it *unstable weight decay* if the weight decay rate is not constant during training, seen in Definition 1. Our empirical analysis suggests that the unstable weight decay problem may undesirably damage performance of popular optimizers.

**2. $L_2$ regularization is unstable weight decay in all optimizers that use Momentum.** Most popular optimizers in modern deep learning libraries use Momentum and $L_2$ regularization at the same time, including SGD (with Momentum). Unfortunately, $L_2$ regularization often damages performance in the presence of Momentum.

**3. Decoupled weight decay is unstable weight decay in adaptive gradient methods.** All adaptive gradient methods used $L_2$ regularization as weight decay until Loshchilov & Hutter (2018) proposed decoupled weight decay. However, decoupled weight decay only solves part of the unstable weight decay problem. Decoupled weight decay is still unstable in the presence of Adaptive Learning Rate.

**4. Always make weight decay rates stable.** We proposed the *stable weight decay* (SWD) method which applies a bias correction factor on decoupled weight decay to make weight decay more stable during training. SWD makes significant improvements over $L_2$ regularization and decouple weight decay in our experiments. We display the test performance in Table 1. The Adam with SWD (AdamS) is displayed in Algorithm 4.

## 2 A DYNAMICAL PERSPECTIVE ON WEIGHT DECAY

In this section, we study how weight decay affects learning dynamics and how to quantitatively measure the effect of weight decay from a viewpoint of learning dynamics. We also reveal the hidden weight decay problem in SGD with Momentum.

**A dynamical perspective on weight decay.** We first present a new theoretical tool for understanding the effect of weight decay. The vanilla weight decay described by Hanson & Pratt (1989) is given by Equation 1. A more popular implementation for vanilla SGD in modern deep learning libraries is given by

$$\theta_t = (1 - \eta\lambda)\theta_{t-1} - \eta\frac{\partial L(\theta_{t-1})}{\partial \theta}, \tag{2}$$

where we denote the training loss of one minibatch as $L(\theta)$ and weight decay should be coupled with the learning rate.

We define new coordinates $w_t \equiv \theta_t(1 - \eta\lambda)^{-t}$, which is an iteration-dependent rescaled system of $\theta$. In the system of $w$, we may define the loss function of $w$ as $L^w(w_t) \equiv L((1 - \eta\lambda)^t w_t) = L(\theta_t)$.

Then, we may rewrite Equation 2 as

$$w_t = w_{t-1} - (1 - \eta\lambda)^{-t}\eta\frac{\partial L(\theta_{t-1})}{\partial \theta} = w_{t-1} - (1 - \eta\lambda)^{-2t+1}\eta\frac{\partial L^w(w_{t-1})}{\partial w}. \qquad (3)$$

Equation 3 describes learning dynamics in the rescaled coordinates of $w$. The equivalence of Equation 2 and Equation 3 suggests that learning dynamics with weight decay in the original coordinates of $\theta$ is equivalent to learning dynamics without weight decay in the rescaled coordinates of $w$. Thus, the effect of weight decay can be interpreted as flattening the loss landscape of $\theta$ by a factor of $(1 - \eta\lambda)$ per iteration and increase the learning rate by a factor of $(1 - \eta\lambda)^{-2}$ per iteration.

**The weight decay rate and the total weight decay effect.** In dynamics of SGD, the learning rate $\eta$ may be interpreted as unit stepsize or unit dynamical time. The dynamical time of deep learning dynamics at $t$-th iteration can be defined as the cumulative sum of stepsizes: $\tau = t\eta$. Thus, for each time step as $d\tau = \eta$, it is $(1 - \lambda)$ that decides the rescaling rate of the dynamical system $w$ per unit dynamical time. We define the *weight decay rate* as the rescaling rate of the dynamical system $w$ per unit dynamical time:

$$R = 1 - \lambda, \qquad (4)$$

which indicates the rescaling rate of the coordinates $w$ per unit stepsize or unit dynamical time. We define the *total weight decay effect* after $t$ iterations as

$$\rho_t = (1 - \eta\lambda)^t, \qquad (5)$$

which indicates the total flattened ratio of the coordinates $w$ after $t$ iterations. We naturally have $\rho_t = R^\tau$ in the continuous-time limit or small $\eta$ limit. Based on the weight decay rate, we define the key concept, stable weight decay, as Definition 1.

**Definition 1** (Stable Weight Decay). *The weight decay is stable if the weight decay rate is constant during training. Otherwise, we define it unstable weight decay.*

Suppose there exists an optimal weight decay rate for training of a deep network. Then, we should fine-tune $\lambda$ to have a proper weight decay rate and keep the the weight decay rate approximately equal to the optimal weight decay rate during training. Thus, we conjecture that stable weight decay should be a desirable property for deep learning. In this paper, we conducted comprehensive empirical analysis to verify the advantage of stable weight decay.

**Vanilla weight decay is unstable weight decay in the presence of learning rate schedulers.** We first take vanilla SGD as the studied example, where no Momentum is involved. It is easy to see that vanilla SGD with $L_2$ regularization $\frac{\lambda}{2}\|\theta\|^2$ is also given by Equation 2. Suppose the learning rate is fixed in the whole training procedure. Then Equation 2 will be identical to Equation 1 if we simply choose $\lambda_0 = \eta\lambda$. However, learning rate decay is quite important for training of deep neural networks. So Equation 2 is not identical to Equation 1 in practice.

Which implementation is better? Based on the measure of the weight decay rate, we propose Proposition 1. We leave all proofs in Appendix A.

**Proposition 1.** *Suppose learning dynamics is governed by Equation 1 and the learning rate at $t$-th step is $\eta_t$. Then, the weight decay rate at $t$-th step is given by $R = 1 - \frac{\lambda_0}{\eta_t}$.*

**Corollary 2.0.1.** *Suppose all conditions of Proposition 1 hold. Then, Equation-1-based weight decay is unstable weight decay in the presence of learning rate schedulers.*

Based on Definition 1 and Proposition 1, we naturally obtain Corollary 2.0.1. We argue that Equation-2-based weight decay is better than Equation-1-based weight decay, because Equation 2 can make the weight decay rate stable. In Figure 1, we empirically verified that the popular Equation-2-based weight decay indeed outperforms the vanilla implementation in Equation 1. Fortunately, the Equation-2-based implementation has been widely accepted by modern deep learning libraries.

$L_2$ **regularization is unstable weight decay in the presence of Momentum.** Our previous analysis revealed that $L_2$ regularization is stable weight decay for vanilla SGD. It is also commonly believed that $L_2$ regularization is an optimal weight decay implementation when Adaptive Learning Rate is not involved. Almost all deep learning libraries directly use $L_2$ regularization as the default weight decay implementation. However, we discover that $L_2$ regularization is not identical to weight decay and often harms performance slightly when Momentum is involved.

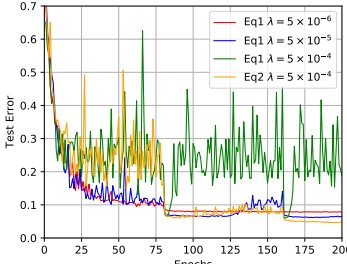 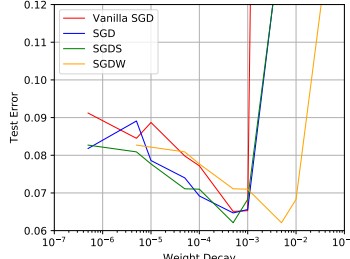

Figure 1: We compared Equation-1-based weight decay and Equation-2-based weight decay by training ResNet18 on CIFAR-10 via vanilla SGD. We divided the learning rate by 10 at the epoch of 80 and 160, which is a popular learning rate scheduler. No matter how we choose $\lambda$ for Equation-1-based weight decay, Equation-2-based weight decay shows better test performance. It demonstrates that the form $-\eta\lambda\theta$ is a better weight decay implementation than $-\lambda\theta$ in the presence of learning rate schedulers.

Figure 2: We compare the generalization of Vanilla SGD, SGD, SGDW, and SGDS under various weight decay hyperparameters by training VGG16 on CIFAR-10. The optimal performance of SGDS/SGDW is better than the widely used Vanilla SGD and SGD. For Vanilla SGD, SGD, and SGDS, we may choose $\lambda_{L_2} = \lambda_S = 0.0005$ to maintain the optimal performance. But we have to re-tune $\lambda_W = 0.005$ for SGDW. Hyperparameter Setting: $\beta_1 = 0$ for Vanilla SGD; $\beta_1 = 0.9$ for SGD, SGDW, and SGDS. We repeat each simulation for three runs. A similar experimental result for ResNet18 is presented in Appendix C.

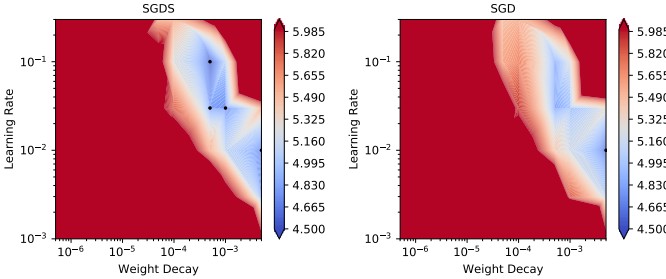

Figure 3: The test errors of ResNet18 on CIFAR-10. SGDS has a deeper blue basin near dark points ($\leq 4.83\%$). The optimal choices of $\eta$ and $\lambda$ are very close for SGDS and SGD.

We take Heavy Ball Method (HBM) (Zavriev & Kostyuk, 1993), which only uses fixed momentum inertia and dampening coefficients, as the studied example in the presence of Momentum, as HBM-style Momentum methods are widely used by many popular optimizers. SGD implemented in PyTorch (Paszke et al., 2019) is actually HBM with default hyperparameters. SGD in TensorFlow (Abadi et al., 2016) follows a slightly different style of Momentum, which uses the learning rate inside the updating rule of Momentum, seen in Appendix E. In this paper, we use the popular implementation of SGD in PyTorch, displayed in Algorithm 1, as the base implementation, as it follows the style of HBM. Note that our theoretical results can be very easily generalized to the SGD implementation in TensorFlow. We write HBM with $L_2$ regularization as

$$\begin{cases} m_t = \beta_1 m_{t-1} + \beta_3(g_t + \lambda_{L_2}\theta_{t-1}) \\ \theta_t = \theta_{t-1} - \eta m_t, \end{cases} \tag{6}$$

where $\beta_1$ and $\beta_3$ are the hyperparameters. In the common setting of SGD with Momentum, $\beta_1 = 0.9$ and $\beta_3 = 1$. We propose Proposition 2 which indicates that $L_2$ regularization is unstable in the presence of Momentum.

**Proposition 2.** *Suppose learning dynamics is governed by SGD with $L_2$ regularization and $\beta_1 > 0$. Then, the followings hold:*
*(1) $L_2$ regularization is not identical to weight decay;*
*(2) the weight decay rate of $L_2$ regularization is not constant during training.*

**Fixing weight decay in Momentum.** For obtaining a stable weight decay rate, we propose the SWD correction for HBM as

$$
\begin{cases}
m_t = \beta_1 m_{t-1} + \beta_3 g_t \\
\theta_t = \left[ 1 - \frac{\beta_3(1-\beta_1^t)}{1-\beta_1} \eta \lambda \right] \theta_{t-1} - \eta m_t,
\end{cases}
\tag{7}
$$

where $\frac{\beta_3(1-\beta_1^t)}{1-\beta_1}\eta$ is the effective learning rate in HBM, as $m_t \approx \frac{\beta_3(1-\beta_1^t)}{1-\beta_1}\mathbb{E}[g_t]$. As $(1-\beta_1^t)$ converges to 1 soon, we use the simplified $\frac{\beta_3}{1-\beta_1}\eta$ as the bias correction in practice, where a relatively large weight decay rate in the first dozen iterations can work like a model parameter initialization strategy. It is easy to see that the bias correction factor $\frac{\beta_3}{1-\beta_1}$ for weight decay is exactly the difference between our stable weight decay and decoupled weight decay suggested by Loshchilov & Hutter (2018). The pseudocode of SGD with SWD (SGDS) is displayed in Algorithm 2. It may also be called HBM with SWD (HBMS).

---

**Algorithm 1:** SGD/SGDW (HBM/HBMW)

$g_t = \nabla L(\theta_{t-1}) + \lambda\theta_{t-1}$;
$m_t = \beta_1 m_{t-1} + \beta_3 g_t$;
$\theta_t = \theta_{t-1} - \eta m_t - \eta\lambda\theta_{t-1}$;

---

**Algorithm 2:** SGDS (HBMS)

$g_t = \nabla L(\theta_{t-1})$;
$m_t = \beta_1 m_{t-1} + \beta_3 g_t$;
$\theta_t = \theta_{t-1} - \eta m_t - \frac{\beta_3}{1-\beta_1}\eta\lambda\theta_{t-1}$;

---

Now, we have three kinds of weight decay hyperparameters. We use $\lambda_{L_2}$, $\lambda_W$, and $\lambda_S$ denote the $L_2$ regularization hyperparameter, the decoupled weight decay hyperparameter, and the stable weight decay hyperparameter, respectively.

**Empirical Analysis on SGDS.** We empirically verified that the optimal performance of SGDS/SGDW is often better than the widely used vanilla SGD and SGD, seen in Figures 2 and 10. We leave more empirical results of SGDS in Appendix C, such as Table 2. Particularly, we report the test performance of SGDS and SGD for ResNet18 on CIFAR-10 under various learning rates and weight decay in Figure 3. SGDS has a deeper blue basin than SGD.

It is not well-known that SGDW often outperforms SGD slightly. We believe it is mainly because people rarely know re-tuning $\lambda_W$ based on $\frac{\beta_3}{1-\beta_1}$ is necessary for maintaining good performance when people switch from SGD to SGDW. As the effective learning rate of SGD is $\frac{\beta_3}{1-\beta_1}\eta$, the weight decay rate of SGDW is actually $R = \frac{1-\beta_1}{\beta_3}\lambda_W$ rather than $\lambda_W$. If we use different settings of $\beta_1$ and $\beta_3$ in decoupled weight decay, we will undesirably change the weight decay rate $R$ unless we re-tune $\lambda_W$. However, people usually directly let $\lambda_W = \lambda_{L_2}$ for SGDW in practice. Figure 2 shows that, the optimal $\lambda_{L_2}$ and $\lambda_S$ in SGD ( with $\beta_1 = 0.9$ and $\beta_3 = 1$) are both 0.0005, while the optimal $\lambda_W$ is 0.005 instead. The optimal $\lambda_{L_2}$ and $\lambda_S$ are almost same, while the optimal $\lambda_W$ is quite different. Thus, the advantage of SGDS over SGDW can save us from re-tuning the weight decay hyperparameter.

## 3 FIXING WEIGHT DECAY IN ADAPTIVE GRADIENT METHODS

In this section, we study weight decay in dynamics of adaptive gradient methods. Loshchilov & Hutter (2018) first pointed that, when the learning rate is adaptive, the commonly used $L_2$ regularization is not identical to weight decay.

---

**Algorithm 3:** Adam/AdamW

$g_t = \nabla L(\theta_{t-1}) + \lambda\theta_{t-1}$;
$m_t = \beta_1 m_{t-1} + (1-\beta_1)g_t$;
$v_t = \beta_2 v_{t-1} + (1-\beta_2)g_t^2$;
$\hat{m}_t = \frac{m_t}{1-\beta_1^t}$;
$\hat{v}_t = \frac{v_t}{1-\beta_2^t}$;
$\theta_t = \theta_{t-1} - \frac{\eta}{\sqrt{\hat{v}_t}+\epsilon}\hat{m}_t - \eta\lambda\theta_{t-1}$;

---

**Algorithm 4:** AdamS

$g_t = \nabla L(\theta_{t-1})$;
$m_t = \beta_1 m_{t-1} + (1-\beta_1)g_t$;
$v_t = \beta_2 v_{t-1} + (1-\beta_2)g_t^2$;
$\hat{m}_t = \frac{m_t}{1-\beta_1^t}$;
$\hat{v}_t = \frac{v_t}{1-\beta_2^t}$;
$\bar{v}_t = mean(\hat{v}_t)$;
$\theta_t = \theta_{t-1} - \frac{\eta}{\sqrt{\hat{v}_t}+\epsilon}\hat{m}_t - \frac{\eta}{\sqrt{\bar{v}_t}}\lambda\theta_{t-1}$;

---

**Decoupled weight decay is unstable weight decay in adaptive gradient methods.** In the following analysis, we ignore the effect of Momentum and focus on the effect of Adaptive Learning Rate. Thus, AdamW dynamics can be written as

$$\theta_t = (1 - \eta\lambda)\theta_{t-1} - \eta v_t^{-\frac{1}{2}} \frac{\partial L(\theta_{t-1})}{\partial \theta}, \tag{8}$$

where $v_t$ is the exponential moving average of the squared gradients in Algorithm 3 and the power notation of a vector means the element-wise power of the vector. We interpret $\eta v_t^{-\frac{1}{2}}$ as the effective learning rate of adaptive gradient methods. We clearly see that decoupled weight decay uses the vanilla learning rate rather than the effective learning rate. Thus, the weight decay rate is anisotropic in AdamW. We propose Proposition 3.

**Proposition 3.** *Suppose learning dynamics is governed by AdamW. Then, the weight decay rate $R_W$ is given by*

$$R_W = \mathbf{1} - \lambda v^{\frac{1}{2}}, \tag{9}$$

*where $\mathbf{1}$ is the all-ones vector. Thus, decoupled weight decay is unstable weight decay in adaptive gradient methods.*

We also note that a good property of decoupled weight decay is that the total weight decay effect after $t$ iterations is still $\rho_t = (1 - \eta\lambda)^t$. In summary, the advantage of AdamW is that the total weight effect $\rho$ after training is isotropic and does not depend on $v$, while the disadvantage of AdamW is that $v$ makes the weight decay rate anisotropic and unstable during training.

**Fixing weight decay in adaptive gradient methods.** It is easy to fix weight decay by using the following bias-corrected weight decay:

$$\theta_t = (\mathbf{1} - \eta v_t^{-\frac{1}{2}}\lambda)\theta_{t-1} - \eta v_t^{-\frac{1}{2}} \frac{\partial L(\theta_{t-1})}{\partial \theta}, \tag{10}$$

where the weight decay rate $R = (1 - \lambda)$. Without losing generality, we first assume the dimensionality of $\theta$ is one. The bias correction factor $v_t^{-\frac{1}{2}}$ make the weight decay rate ideally stable during training in one-dimensional dynamics.

However, if we still use the element-wise bias correction factor $v_t^{-\frac{1}{2}}$ in multi-dimensional dynamics, the total weight decay effect $\rho$ will become highly different along different dimensions: $\rho_t = (1 - \lambda)^{\sum_{k=1}^{t} \eta v_k^{-\frac{1}{2}}}$. The anisotropic total weight decay effect is undesirable. Unfortunately, there is no simple solution to achieving both an ideally stable weight decay rate and an isotropic total weight decay effect at the same time for adaptive gradient methods. This may be an internal fault of all optimizers that use Adaptive Learning Rate.

Although there is no ideal stable weight decay for Adam, it is possible to apply an isotropic bias correction term to implement relatively more stable weight decay. We propose the updating rule in presence of Adaptive Learning Rate and SWD as:

$$\theta_t = (\mathbf{1} - \eta\bar{v}_t^{-\frac{1}{2}}\lambda)\theta_{t-1} - \eta v_t^{-\frac{1}{2}} \frac{\partial L(\theta_{t-1})}{\partial \theta}, \tag{11}$$

where $\bar{v}_t$ is the mean of all elements of the vector $v_t$. This is exactly the updating rule of AdamS, displayed in Algorithm 4. Here $\eta\bar{v}_t^{-\frac{1}{2}}$ is for approximating the effective learning rate. As $\bar{v}_t^{-\frac{1}{2}}$ is isotropic, the total weight decay effect is still same along different dimensions. The weight decay rate $R_S$ of AdamS dynamics is given by

$$R_S = \mathbf{1} - \lambda\bar{v}_t^{-\frac{1}{2}}v_t^{\frac{1}{2}}. \tag{12}$$

Finally, we propose Propositions 4 and 5 for quantitatively measuring the magnitude of the weight decay rate of AdamS and AdamW. As $L_2$ norm is the most popular measure for the magnitude of vectors, we measure the stability of AdamW and AdamS in terms of $L_2$ norms of the coefficients of $\lambda$ in the corresponding weight decay rate.

**Proposition 4.** *For the weight decay rate of AdamW in Equation 9, the $L_2$ norm of the vector coefficient of $\lambda$ is equal to the sum of all elements in $v_t$, which is highly varying during training.*

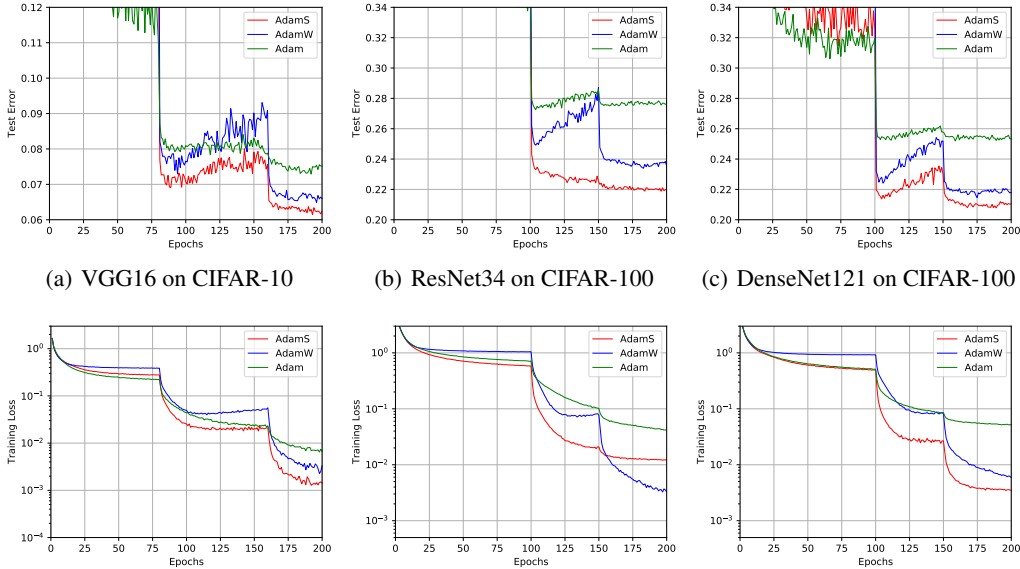

Figure 4: The learning curves of AdamS, AdamW, and Adam on CIFAR-10 and CIFAR-100. Top Row: Test curves. Bottom Row: Training curves. AdamS shows significantly better generalization than AdamW and Adam.

**Proposition 5.** *For the weight decay rate of AdamS in Equation 12, the $L_2$ norm of the vector coefficient of $\lambda$ is equal to the space dimensionality, which is constant during training.*

Based on Propositions 4 and 5, we argue that SWD is more stable than decoupled weight decay in adaptive gradient methods in the sense of the magnitude.

## 4 EMPIRICAL ANALYSIS AND DISCUSSION

In this section, we empirically study how the SWD-corrected optimizers are compared with conventional optimizers. We choose Adam as the base optimizer, and train popular deep models, including ResNet18/ResNet34 (He et al., 2016), VGG16 (Simonyan & Zisserman, 2014), DenseNet121 (Huang et al., 2017), GoogLeNet (Szegedy et al., 2015), and Long Short-Term Memory (LSTM) (Hochreiter & Schmidhuber, 1997), on CIFAR-10/CIFAR-100 (Krizhevsky et al., 2009) and Penn TreeBank (Marcus et al., 1993). The implementation details can be found in Appendix B.

**Empirical results.** Figure 4 shows the learning curves of AdamS, AdamW, and Adam on several benchmarks. In our experiments, AdamS always leads to lower test errors, and sometimes leads to lower training losses. However, Figure 5 shows that, even if with similar or higher training losses, AdamS still generalizes significantly better than AdamW, Adam, and popular Adam variants. Figure 6 displays the learning curves of all adaptive gradient methods. The test performance of other models can be found in Table 1. Simply fixing weight decay in Adam by SWD even outperforms recent Adam variants. Figure 7 further demonstrates that AdamS consistently outperforms Adam and AdamW under various weight decay hyperparameters. According to Figure 7, we also notice that the optimal decoupled weight decay hyperparameter in AdamW can be very different from $L_2$ regularization and stable weight decay, but the corresponding optimal weight decay rates of three weight decay implementations are often similar. Finally, AdamS also outperforms AdamW and Adam significantly on the Language Modeling experiment, seen in Figure 8. We leave the experiments with cosine annealing schedulers and warm restarts (Loshchilov & Hutter, 2016) in Appendix F, which also support that AdamS yields superior test performance.

**Popular Adam variants often generalize worse than SGD.** Some papers (Luo et al., 2019; Chen & Gu, 2018) argued that their proposed Adam variants may generalize as well as SGD. However, we found that this argument is contracted with our comparative experimental results, such as Table

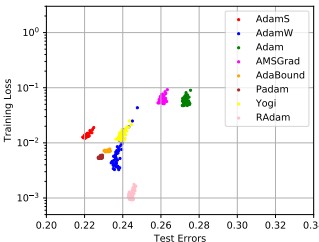 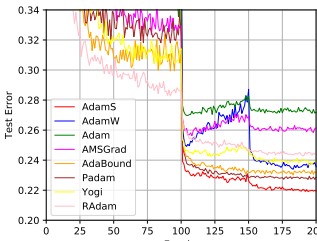 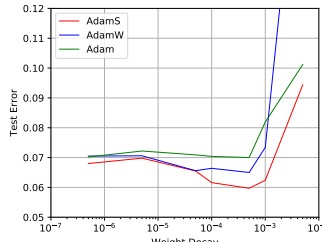

Figure 5: The scatter plot of training losses and test errors during final 40 epochs of training ResNet34 on CIFAR-100. Even with similar or higher training losses, AdamS still generalizes better than other Adam variants. We leave the scatter plot on CIFAR-10 in Appendix D.

Figure 6: The learning curves of all adaptive gradient methods by training ResNet34 on CIFAR-100. AdamS outperforms other Adam variants. The test performance of other models can be found in Table 1.

Figure 7: The test errors of VGG16 on CIFAR-10 with various weight decay rates. The displayed weight decay value of AdamW has been rescaled by the bias correction factor $\approx$ 0.001. A similar experimental result for ResNet34 is presented in Appendix D.

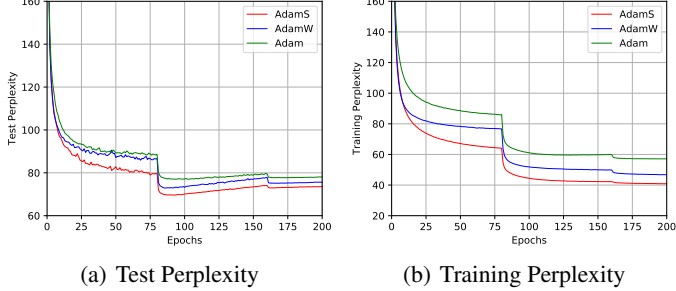

(a) Test Perplexity  (b) Training Perplexity

Figure 8: Language Modeling. The learning curves of AdamS, AdamW, and Adam for LSTM on Penn TreeBank. AdamS has better test performance than AdamW and Adam for LSTM. The optimal test perplexity of AdamS, AdamW, and Adam are 69.90, 72.88, and 77.01, respectively. Note that the lower perplexity is better.

1. In our empirical analysis, these advanced Adam variants may narrow but not completely close the generalization gap between adaptive gradient methods and SGD. SGD with a fair weight decay hyperparameter as the baseline performance usually generalizes better than recent adaptive gradient methods. The main problem may lie in weight decay. SGD with weight decay $\lambda = 0.0001$, a common setting in related papers, is often not a good baseline, as $\lambda = 0.0005$ often shows better generalization on CIFAR-10 and CIFAR-100, seen in Figures 2, 10 and 7. We also conduct comparative experiments with $\lambda = 0.0001$, seen in Table 3 of Appendix D. Under the setting $\lambda = 0.0001$, while some existing Adam variants may outperform SGD sometimes due to the lower baseline performance of SGD, AdamS shows superior test performance. For example, for ResNet18 on CIFAR-10, the test error of AdamS is lower than SGD by nearly one point and no other Adam variant may compare with AdamS.

**Two metrics for weight decay.** The main effect of weight decay for deep learning dynamics can be reflected by these two metrics: the weight decay rate $R$ and the total weight decay effect $\rho$. We summary two principles about weight decay critically related to performance. First, we need to make the weight decay rate as stable as possible during training. Second, we need choose the optimal weight decay hyperparameter to maintain the optimal total weight decay effect. These two principles lead to two linear scaling rules.

**Two linear scaling rules.** We report two kinds of *linear scaling rules* of weight decay and learning rates for selecting the weight decay hyperparameter $\lambda$. First, as Proposition 1 and Figure 1 indicate, the learning rate scheduler should be applied to both the learning rate and weight decay in the form of $-\eta_t \lambda \theta_{t-1}$ during training. Second, the initial learning rate and the weight decay hyperparameter $\lambda$

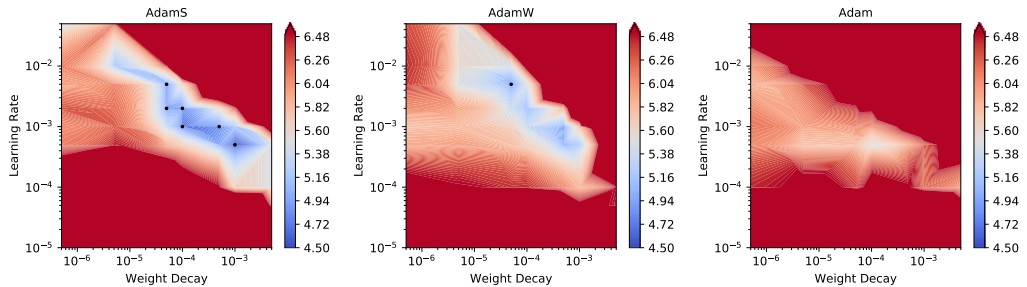

Figure 9: The test errors of ResNet18 on CIFAR-10. AdamS has a much deeper and wider basin near dark points ($\leq 4.9\%$). The optimal test error of AdamS, AdamW, and Adam are $4.52\%$, $4.90\%$, and $5.49\%$, respectively. The displayed weight decay value of AdamW has been rescaled by the bias correction factor $\approx 0.001$.

are linearly coupled for the total weight decay effect. The expression of the total weight decay effect in Equation 5 mainly depends on the weight decay hyperparameter $\lambda$, the learning rate $\eta$, and the number of iterations $t$. Usually, people do not fine tune the number of iterations $t$. Given a fixed $t$ and the optimal weight decay rate $R$, the total weight decay effect $\rho$ approximately depends on $\eta\lambda$. The second linear scaling rule indicates that, in the procedure of fine-tuning hyperparameters, we should try larger $\lambda$ for smaller initial learning rates to maintain the optimal total weight decay effect. We observe that the blue basin of AdamS and AdamW in Figure 9 both approximately reflect the linear scaling rule, which was also reported by Van Laarhoven (2017); Loshchilov & Hutter (2018); Li et al. (2020). We point that its theoretical mechanism closely relates to the total weight decay effect.

**A novel interpretation of weight decay.** The effect of weight decay can be interpreted as iteration-wisely flattening the loss landscape and increasing the learning rate at the same time. Thus, it is not surprising that increasing weight decay and increasing learning rate have some similar influence on optimization performance (Van Laarhoven, 2017; Zhang et al., 2018; Hoffer et al., 2018). However, simply increasing learning rates cannot replace weight decay in practice. For weight decay, increasing the learning rate by a factor of $(1 - \eta\lambda)^{-2}$ is iteration-wise during training, and flattening the loss landscape by a factor of $(1 - \eta\lambda)$ per iteration must happen at the same time. This interpretation of weight decay actually controls the adaptivity of learning rates in learning dynamics, as the total weight decay effect $\rho$ is isotropic along different dimensions. This may explain why stable weight decay can effectively fix the ill-conditioned flat minima selection of adaptive gradient methods revealed by Xie et al. (2020b) from a diffusion perspective (Xie et al., 2020a).

## 5 CONCLUSION

While Loshchilov & Hutter (2018) discovered that weight decay should be decoupled from the gradients, we further discovered that weight decay should be coupled with the effective learning rate. We proposed the SWD method which applies the bias correction on decoupled weight decay to make the weight decay rate more stable during training. The empirical results demonstrate that SWD makes significant improvements over $L_2$ regularization and decoupled weight decay (when they are different). Particularly, we observe that the advantage of existing Adam variants is very weak compared with the performance improvement by simply fixing weight decay. The standard Adam with SWD, with no extra hyperparameter, usually outperforms complex Adam variants, such as Padam and AdaBound, which have more hyperparameters. Although our analysis mainly focused on Adam, SWD can be easily combined with other Adam variants, too. The proposed principle and method of stable weight decay expands our understanding towards the role of weight decay in deep learning. It will be interesting to further investigate the theory for the weight decay rate in future.

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

## A PROOFS

### A.1 PROOF OF PROPOSITION 1

*Proof.* In the presence of learning rate schedulers, we have the updating rule of Equation-1-based weight decay as

$$\theta_t = (1 - \lambda_0)\theta_{t-1} - \eta_t g_t. \tag{13}$$

According to the definition of the weight decay rate, we have $R = 1 - \frac{\lambda_0}{\eta_t}$ at $t$-th step.

Obviously, the weight decay rate will change with learning rate schedulers.

The proof is now complete. □

## A.2 PROOF OF PROPOSITION 2

*Proof.* We first show how $L_2$ regularization and weight decay are different in the presence of Momentum.

We may rewrite HBM with $L_2$ regularization as

$$
\begin{cases}
m_t = \beta_1 m_{t-1} + \beta_3 g_t \\
\theta_t = \theta_{t-1} - \eta m_t - \eta \lambda_{L_2} \beta_3 \prod_{k=1}^t \beta_1^{k-1} \theta_{t-k} = \theta_{t-1} - \eta m_t - \eta \lambda_{L_2} \beta_3 \frac{1 - \beta_1^t}{1 - \beta_1} \langle \theta_{t-1} \rangle,
\end{cases}
\tag{14}
$$

where we denote the exponential moving average of past $\theta$ as $\langle \theta_{t-1} \rangle = \frac{1 - \beta_1}{1 - \beta_1^t} \prod_{k=1}^t \beta_1^{k-1} \theta_{t-k}$.

Although $\langle \theta_{t-1} \rangle$ is an approximated value of $\theta_{t-1}$, $L_2$ regularization use $\langle \theta_{t-1} \rangle$ instead of the standard $\theta_{t-1}$ in weight decay introduces undesirable noise into learning dynamics of SGD with $L_2$ regularization.

As $\langle \theta_{t-1} \rangle \neq \theta_{t-1}$, obviously, we may not recover weight decay in Equation 2 from $L_2$ regularization. This naturally means that $L_2$ is not identical to weight decay.

The weight decay rate of SGD with $L_2$ is given by

$$
R = \mathbf{1} - \lambda_{L_2} \beta_3 \frac{1 - \beta_1^t}{1 - \beta_1} \langle \theta_{t-1} \rangle \theta_{t-1}^{-1},
\tag{15}
$$

where $\mathbf{1}$ is the all-ones vector and $\theta_{t-1}^{-1}$ is the element-wise inverse of $\theta_{t-1}$.

The difference between $\theta_{t-1}$ and $\langle \theta_{t-1} \rangle$ also indicates that the weight decay rate cannot be constant during training.

The proof is now complete. □

## A.3 PROOF OF PROPOSITION 3

*Proof.* According to the definition of the weight decay rate, the weight decay rate of AdamW dynamics is

$$
R_W = \mathbf{1} - \lambda v^{\frac{1}{2}},
\tag{16}
$$

where $\mathbf{1}$ is the all-ones vector.

As $v^{-\frac{1}{2}}$ is adaptive, $R_W$ is not constant during training.

The proof is now complete. □

## A.4 PROOF OF PROPOSITIONS 4 AND 5

*Proof.* By Equation 9 and Equation 12, we have the weight decay rate of AdamW as

$$
R_W = \mathbf{1} - \lambda v^{\frac{1}{2}},
\tag{17}
$$

and the weight decay rate of AdamS as

$$
R_S = \mathbf{1} - \lambda \bar{v}_t^{-\frac{1}{2}} v_t^{\frac{1}{2}}.
\tag{18}
$$

We note that $\lambda$ in Equation 12 (AdamS) has a vector coefficient as $\bar{v}_t^{-\frac{1}{2}} v_t^{\frac{1}{2}}$, while $\lambda$ in Equation 9 (AdamW) has a vector coefficient as $v_t^{\frac{1}{2}}$.

It is easy to compute the $L_2$ norms of two vector coefficients: $\|\bar{v}_t^{-\frac{1}{2}} v_t^{\frac{1}{2}}\|_2$ equals the dimensionality $n$ in AdamS dynamics and $\|v_t^{\frac{1}{2}}\|_2$ equals the sum of all elements in $v_t$ in AdamW dynamics.

It means $\lambda$ in AdamS has a magnitude-constant coefficient, while $\lambda$ in AdamW has a magnitude-varying coefficient.

The proof is now complete. □

Table 2: Test performance comparison of Adai, AdaiS, SGD, and SGDS. Stable/Decoupled Weight Decay often outperform $L_2$ regularization for optimizers involving in momentum. We report the mean and the standard deviations (as the subscripts) of the optimal test errors computed over three runs of each experiment.

| DATASET | MODEL | ADAIS | ADAI | SGDS | SGD |
|---------|-------|-------|------|------|-----|
| CIFAR-10 | RESNET18 | $\mathbf{4.59}_{0.16}$ | $4.74_{0.14}$ | $4.69_{0.09}$ | $5.01_{0.03}$ |
| | VGG16 | $\mathbf{5.81}_{0.07}$ | $6.00_{0.09}$ | $6.28_{0.07}$ | $6.42_{0.02}$ |
| CIFAR-100 | DENSENET121 | $\mathbf{19.44}_{0.21}$ | $19.59_{0.38}$ | $19.61_{0.26}$ | $19.81_{0.33}$ |
| | GOOGLENET | $\mathbf{20.50}_{0.25}$ | $20.55_{0.32}$ | $20.68_{0.03}$ | $21.21_{0.29}$ |

# B  EXPERIMENTAL DETAILS

## B.1  IMAGE CLASSIFICATION

**Data Preprocessing:** We perform the common per-pixel zero-mean unit-variance normalization, horizontal random flip, and $32 \times 32$ random crops after padding with 4 pixels on each side.

**Hyperparameter Settings:** We select the optimal learning rate for each experiment from $\{0.001, 0.01, 0.1, 1, 10\}$ for non-adaptive gradient methods and use the default learning rate for adaptive gradient methods. In the experiments on CIFAF-10 and CIFAR-100: $\eta = 1$ for Adai; $\eta = 0.1$ for AdaiS; $\eta = 0.1$ for SGD and SGDS; $\eta = 0.001$ for AdamS, Adam, AMSGrad, AdamW, and AdaBound; $\eta = 0.01$ for Padam. For the learning rate schedule, the learning rate is divided by 10 at the epoch of $\{80, 160\}$ for CIFAR-10 and $\{100, 150\}$ for CIFAR-100, respectively. The batch size is set to 128 for both CIFAR-10 and CIFAR-100.

The strength of $L_2$ regularization and SWD is default to $0.0005$ as the baseline. Considering the linear scaling rule, we choose $\lambda_W = \frac{\lambda_{L_2}}{\eta}$. Thus, the weight decay of AdamW uses $\lambda_W = 0.5$ for CIFAR-10 and CIFAR-100. As for SWD in AdaiS, we choose $\lambda_S = 0.005$ for achieve a similar total weight decay effect as SGD. The basic principle of choosing weight decay strength is to let all optimizers have either a similar weight decay rate $R$ or a similar total weight decay effect $\rho$ to the baseline $L_2$ regularization.

We set the momentum hyperparameter $\beta_1 = 0.9$ for SGD and SGDS. As for other optimizer hyperparameters, we apply the default hyperparameter settings directly.

We leave the empirical results with the weight decay setting $\lambda = 0.0001$ in Appendix D.

The source code of AdamS and SGDS in Supplementary Materials is available to the public.

## B.2  LANGUAGE MODELING

We use a classical language model, Long Short-Term Memory (LSTM) (Hochreiter & Schmidhuber, 1997) with 2 layers, 512 embedding dimensions, and 512 hidden dimensions, which has 14 million model parameters and is similar to the "medium LSTM" in Zaremba et al. (2014). Note that our baseline performance is better than the reported baseline performance in Zaremba et al. (2014). The benchmark task is the word-level Penn TreeBank (Marcus et al., 1993). We empirically compared AdamS, AdamW, and Adam under the common and same conditions.

**Hyperparameter Settings.** Batch Size: $B = 20$. BPTT Size: $bptt = 35$. Weight Decay: $\lambda_{L_2} = \lambda_S = 0.00005$, and $\lambda_W = 0.05$. Learning Rate: $\eta = 0.001$. The dropout probability is set to $0.5$. We clipped gradient norm to 1.

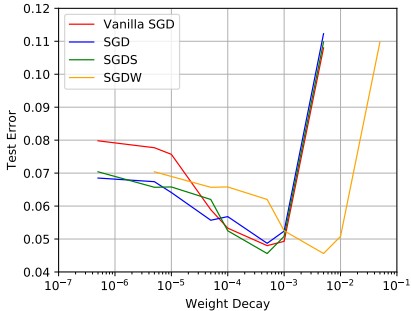

Figure 10: We compare the generalization of Vanilla SGD, SGD, SGDW, and SGDS with various weight decay hyperparameters by training ResNet18 on CIFAR-10. The optimal weight decay rates are near 0.0005 for all three weight implementations. The optimal performance of SGDS/SGDW is better than Vanilla SGD and SGD. For Vanilla SGD, SGD, and SGDS, we may safely choose $\lambda_{L_2} = \lambda_S = 0.0005$. But we have to re-tune $\lambda_W = 0.005$ for SGDW. Hyperparameter Setting: $\beta_1 = 0$ for Vanilla SGD; $\beta_1 = 0.9$ for SGD, SGDW, and SGDS.

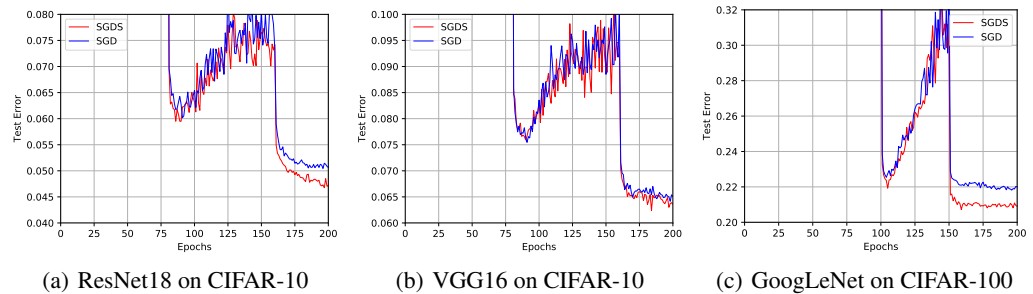

(a) ResNet18 on CIFAR-10      (b) VGG16 on CIFAR-10      (c) GoogLeNet on CIFAR-100

Figure 11: Generalization analysis on SGDS and SGD with $L_2$ regularization. Hyperparameter Setting: $\lambda_S = \lambda_{L_2} = 0.0005$ and $\beta_1 = 0.9$.

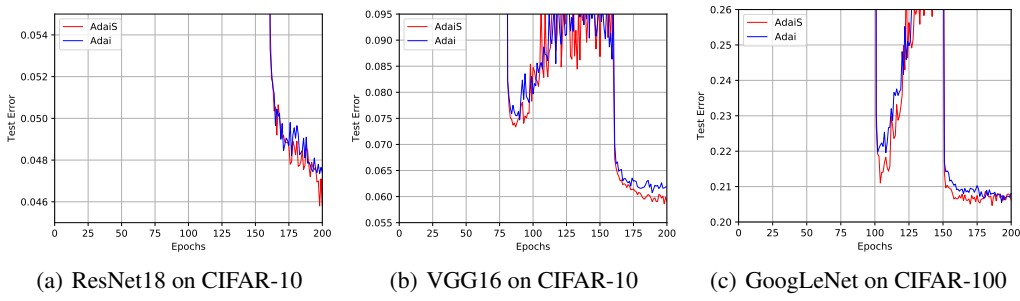

(a) ResNet18 on CIFAR-10      (b) VGG16 on CIFAR-10      (c) GoogLeNet on CIFAR-100

Figure 12: Generalization analysis on AdaiS and Adai with $L_2$ regularization. Hyperparameter Setting: $\lambda_S = \lambda_{L_2} = 0.0005$.

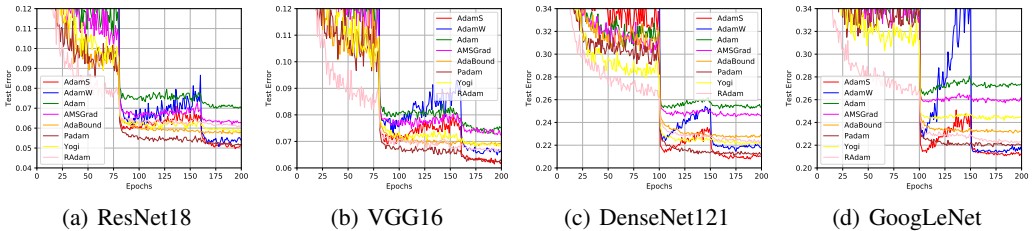

|  |  |  |  |
| :---: | :---: | :---: | :---: |
| (a) ResNet18 | (b) VGG16 | (c) DenseNet121 | (d) GoogLeNet |

Figure 13: The learning curves of adaptive gradient methods.

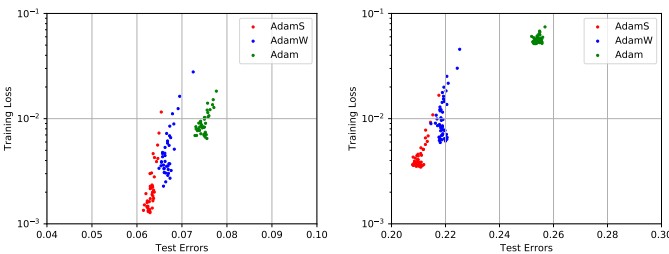

Figure 14: Even if with similar or higher training losses, AdamS still generalizes better than AdamW and Adam. The scatter plot of training losses and test errors during final 50 epochs of training VGG16 on CIFAR-10 and DenseNet121 on CIFAR-100.

## C  SUPPLEMENTARY FIGURES AND RESULTS OF SGD WITH MOMENTUM AND ADAI

We compare SGDS, SGDW, vanilla SGD and SGD under various weight decay hyperparameters by training ResNet18 on CIFAR-10. We observe that the optimal performance of SGDS/SGDW is better than vanilla SGD and SGD in Figure 10. The advantage of SGDS over SGDW is that we do not need to fine-tune the weight decay hyperparameters based on $\frac{\beta_3}{1-\beta_1}$.

We report the learning curves of SGD and SGDS in Figure 11. SGDS compares favorably with SGD.

Based on a diffusion theoretical framework (Xie et al., 2020a), Xie et al. (2020b) proposed Adaptive Inertia Estimation (Adai) that uses adaptive momentum inertia instead of Adaptive Learning Rate to help training. Adaptive inertia can be regarded as an inertia-adaptive variant of HBM. The previous analysis on HBM can be easily generalized to Adai. We display Adai with SWD (AdaiS) in Algorithm 6. We report the learning curves of Adai and AdaiS in Figure 12, which may verify the generalization advantage of AdaiS over Adai.

## D  SUPPLEMENTARY FIGURES AND RESULTS OF ADAPTIVE GRADIENT METHODS

We report the learning curves of all adaptive gradient methods in Figure 13. They shows that vanilla Adam with SWD can outperform other complex variants of Adam.

Figure 14 displays the scatter plot of training losses and test errors during final 40 epochs of training DenseNet121 on CIFAR-100.

Figure 15 displays the test performance of AdamS, AdamW, and Adam under various weight decay hyperparameters of ResNet34 on CIFAR-100.

We train ResNet18 on CIFAR-10 for 900 epochs to explore the performance limit of AdamS, AdamW, and Adam in Figure 16.

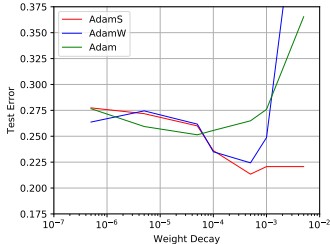

Figure 15: We compare the generalization of Adam, AdamW, and AdamS with various weight decay rates by training ResNet34 on CIFAR-100. The displayed weight decay of AdamW in the figure has been rescaled by the bias correction factor $\approx 0.001$. The optimal test performance of AdamS is significantly better than AdamW and Adam.

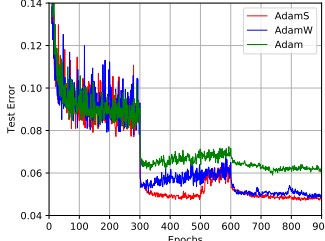

Figure 16: We train ResNet18 on CIFAR-10 for 900 epochs to explore the performance limit of AdamS, AdamW, and Adam. The learning rate is divided by 10 at the epoch of 300 and 600. AdamS achieves the most optimal test error, $4.70\%$.

Table 3 displays the test performance with $\lambda = 0.0001$, which is a common weight decay setting in related papers.

Finally, we present the learning curves for language modeling experiments in Figure 8. As we expect, AdamS outperforms AdamW and Adam again.

## E ADDITIONAL ALGORITHMS

Algorithm 5 is the TensorFlow implementation for SGD.

Algorithm 6 is the implementation of Adai, AdaiW, and AdaiS. As $\frac{\beta_3}{1-\beta_1}$ is always 1 in Adai, AdaiW is identical to AdaiS.

Table 3: Test performance comparison of optimizers with $\lambda_{L_2} = \lambda_S = 0.0001$ and $\lambda_W = 0.1$. AdamS still show better test performance than popular adaptive gradient methods and SGD.

| DATASET | MODEL | SGD | ADAMS | ADAM | AMSGRAD | ADAMW | ADABOUND | PADAM | YOGI | RADAM |
|---------|-------|-----|-------|------|---------|-------|----------|-------|------|-------|
| CIFAR-10 | RESNET18 | 5.58 | **4.69** | 6.08 | 5.72 | 5.33 | 6.87 | 5.83 | 5.43 | 5.81 |
|  | VGG16 | 6.92 | **6.16** | 7.04 | 6.68 | 6.45 | 7.33 | 6.74 | 6.69 | 6.73 |
| CIFAR-100 | DENSENET121 | **20.98** | 21.35 | 24.39 | 22.80 | 22.23 | 24.23 | 22.26 | 22.40 | 22.40 |
|  | GOOGLENET | 21.89 | **21.60** | 24.60 | 24.05 | 21.71 | 25.03 | 26.69 | 22.56 | 22.35 |

We note that the implementation of AMSGrad in Algorithm 7 is the popular implementation in PyTorch. We use the PyTorch implementation in our paper, as it is widely used in practice.

---

**Algorithm 5:** SGD in TensorFlow

$g_t = \nabla L(\theta_{t-1}) + \lambda\theta_{t-1}$;
$m_t = \beta_1 m_{t-1} - \eta g_t$;
$\theta_t = \theta_{t-1} + m_t$;

---

**Algorithm 6:** Adai /AdaiS=AdaiW

$g_t = \nabla L(\theta_{t-1}) + \lambda\theta_{t-1}$;
$v_t = \beta_2 v_{t-1} + (1 - \beta_2)g_t^2$;
$\hat{v}_t = \frac{v_t}{1-\beta_2^t}$;
$\bar{v}_t = mean(\hat{v}_t)$;
$\beta_{1t} = (1 - \beta_0\frac{\hat{v}_t}{\bar{v}_t}).Clip(0, 1 - \epsilon)$;
$m_t = \beta_{1t}m_{t-1} + (1 - \beta_{1t})g_t$;
$\hat{m}_t = \frac{m_t}{1-\prod_{k=1}^{t}\beta_{1k}}$;
$\theta_t = \theta_{t-1} - \eta\hat{m}_t - \eta\lambda\theta_{t-1}$;

---

**Algorithm 7:** AMSGrad/AMSGradW

$g_t = \nabla L(\theta_{t-1}) + \lambda\theta_{t-1}$;
$m_t = \beta_1 m_{t-1} + (1 - \beta_1)g_t$;
$v_t = \beta_2 v_{t-1} + (1 - \beta_2)g_t^2$;
$\hat{m}_t = \frac{m_t}{1-\beta_1^t}$;
$v_{max} = max(v_t, v_{max})$;
$\hat{v}_t = \frac{v_{max}}{1-\beta_2^t}$;
$\theta_t = \theta_{t-1} - \frac{\eta}{\sqrt{\hat{v}_t}+\epsilon}\hat{m}_t - \eta\lambda\theta_{t-1}$;

---

**Algorithm 8:** AMSGradS

$g_t = \nabla L(\theta_{t-1})$;
$m_t = \beta_1 m_{t-1} + (1 - \beta_1)g_t$;
$v_t = \beta_2 v_{t-1} + (1 - \beta_2)g_t^2$;
$\hat{m}_t = \frac{m_t}{1-\beta_1^t}$;
$v_{max} = max(v_t, v_{max})$;
$\hat{v}_t = \frac{v_{max}}{1-\beta_2^t}$;
$\bar{v}_t = mean(\hat{v}_t)$;
$\theta_t = \theta_{t-1} - \frac{\eta}{\sqrt{\hat{v}_t}+\epsilon}\hat{m}_t - \frac{\eta}{\sqrt{\bar{v}_t}}\lambda\theta_{t-1}$;

---

## F EXPERIMENTS WITH COSINE ANNEALING SCHEDULERS AND WARM RESTARTS

In this section, we conducted comparative experiments on AdamS, AdamW, and Adam in the presence of cosine annealing schedulers and warm restarts proposed by Loshchilov & Hutter (2016). We set the learning rate scheduler with a recommended setting of Loshchilov & Hutter (2016): $T_0 = 14$ and $T_{mul} = 2$. The number of total epochs is 210. Thus, we trained each deep network for four runs of warm restarts, where the four runs have 14, 28, 56, and 112 epochs, respectively. Other hyperparameters and details are displayed in Appendix B.

Our experimental results in Figures 17 and 18 suggest that AdamS consistently outperforms AdamW and Adam in the presence of cosine annealing schedulers and warm restarts. It demonstrates that, with various learning rate schedulers, the advantage of SWD may generally hold.

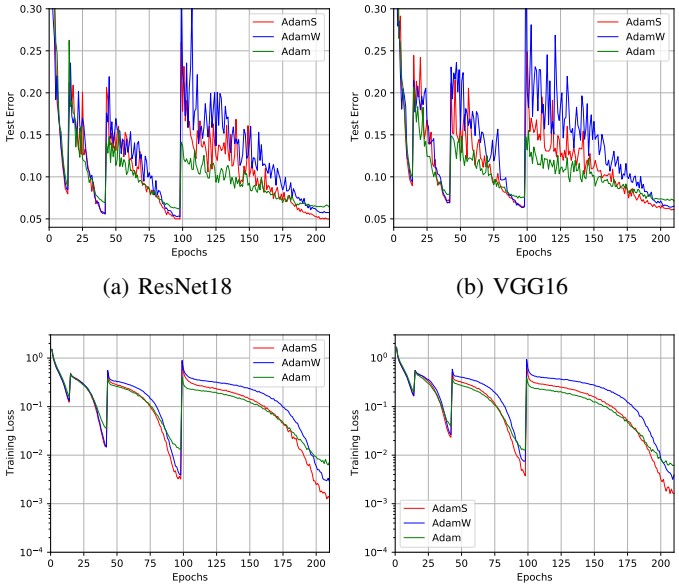

(a) ResNet18         (b) VGG16

Figure 17: The learning curves of ResNet18 and VGG16 on CIFAR-10 with cosine annealing and warm restart schedulers. The weight decay hyperparameter: $\lambda_{L_2} = \lambda_S = 0.0005$ and $\lambda_W = 0.5$. Top Row: Test curves. Bottom Row: Training curves. AdamS yields significantly lower test errors and training losses than AdamW and Adam.

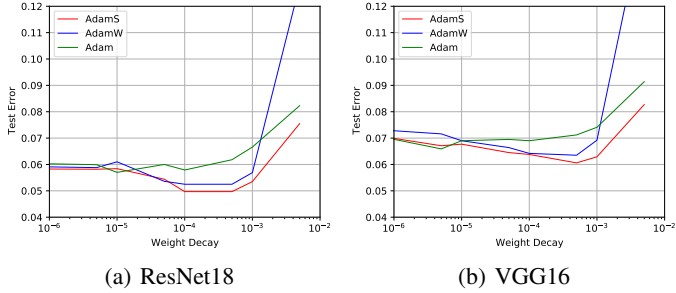

(a) ResNet18         (b) VGG16

Figure 18: The test errors of ResNet18 and VGG16 on CIFAR-10 under various weight decay with cosine annealing and warm restart schedulers. AdamS yields significantly better optimal test performance than AdamW and Adam.

