# OpenReview forum: "Stable Weight Decay Regularization"
_ICLR.cc/2021/Conference — Reject_

### Official Review · AnonReviewer2 · 2020-10-21
**Review of "Stable Weight Decay Regularization"**

**Rating:** 5
**Confidence:** 4

**Review:**

Summary: This paper presents a novel weight decay regularization, stable weight decay by using the bias correction to the decoupled weight decay in adaptive gradient descent optimizer. They empirically found that the L2 regularization the decouple weight decay are unstable weight decay. The proposed stable weight decay can fix this issue in terms of dynamical perspective. Experimental results based on benchmark dataset show that the proposed scheme outperform the popular and advanced optimizers in generalization. Weight decay has been a basic technique in most optimizer and indeed there are not many studies for the effect on the performance. Also, how to obtain the optimal weight decay is still missing.

I think this paper has done a decent investigation on this topic. The overall paper is easy to follow and seems technically sound.

However, one major issue in my mind is that conclusions are only supported by the empirical results, which though look promising. No formal theoretical claims or results have been reported. This hurts the paper in terms of theoretical novelty. Even though some shallow analysis has been presented in the draft, it is not enough.

Moreover, a couple of statements can be arrived at by directly observing existing algorithms, which are not that technical. For example, Statement 1 that says “Equation-1-based weight decay is unstable weight decay in the presence of learning rate scheduler.” can be quickly summarized even from an intuitive sense without any derivation, which makes it trivial. There is no need to give Definition 1 formally for Stable Weight Decay as that doesn’t sound like a definition. In the draft, the authors can directly say constant or time-varying weight decay. Stable doesn’t just mean constant, though I can completely understand what the authors really meant in the paper. Statement 4, in my opinion, is one of key results in the draft. However, the current theoretical analysis is not enough to support this conclusion. More formal theoretical analysis is required.

A minor point, in Eq 3, how did the authors arrive at $-2t-1$ in the superscript of weight decay rate, not $-2t+1$? Additionally, I am a little confused about the statement that the effect of weight decay can be interpreted as flattening the loss landscape of $\theta$ by a factor of $(1-\eta\lambda)$ per iteration and increase the learning rate by a factor of $(1-\eta\lambda)^{-2}$ per iteration. Can the authors put more detail, a bit more of derivation to show this?

***************************************
After reading the rebuttal and considering carefully, I think the authors' response addressed some of issues in my mind so I raised the score. However, in terms of theoretical foundation, the current paper draft is still only marginal, which requires substantial improvement.

---

> ### Author Response · Authors · 2020-11-22
> **Response to Reviewer2**
>
> To Reviewer #2:
>
> We appreciate the reviewer for the hard work and helpful comments.
>
> The main concerns have been duly addressed below.
>
> Q1: One major issue is that conclusions are only supported by the empirical results, which though look promising. No formal theoretical claims or results have been reported. This hurts the paper in terms of theoretical novelty.
>
> A1: Thanks for the comment. We will make more formal theoretical justifications in the next version. We also believe that our current results are promising without formal guarantees. Reference [1] revealed that weight decay should be decoupled from the gradients, and we revealed that weight decay should be coupled with the effective learning rates during training. We both successfully revealed hidden yet important properties about weight decay, although the theoretical guarantee behind the properties is an open problem. The quantitative measures which we proposed may be a contribution in this direction. The quantitative measures which we proposed may be a step forward. We will try to study the theoretical guarantee in future.
>
> Q2: A couple of statements can be arrived at by directly observing existing algorithms, which are not that technical or seem trivial.
>
> A2: We respectfully argue that some statements in Section 2 may seem obvious only if we analyze the weight decay rate, which is originally proposed in our paper. Moreover, the simplicity does not mean these statements are not important. The statements pointed out several specific conditions when $L_{2}$ regularization, Decoupled Weight Decay, and Stable Weight Decay are different. Note that the conclusion in Reference [1] that $L_{2}$ regularization is not identical to weight decay is even more trivial and obvious. However, people had missed the empirical advantage of Decoupled Weight Decay over $L_{2}$ regularization in adaptive gradient methods for many years before Reference [1].
>
>
> Q3: A minor point, in Eq 3, how did the authors arrive at $-2t-1$ in the superscript of weight decay rate, not $-2t+1$?
>
> A3: Thank you very much for pointing out the typo. It should be $-2t+1$. We will correct it.
>
>
> Q4: Statement 4, in my opinion, is one of key results in the draft. However, the current theoretical analysis is not enough to support this conclusion. More formal theoretical analysis is required.
>
> A4: Thanks for the suggestion. We will provide more formal analysis in the next version. We plan to theoretically analyze the magnitude ($L_{2}$ norm) of the vector coefficients of $\lambda$ in the weight decay rate of AdamS/AdamW. We prove that $\lambda$ in the weight decay rate of AdamS has a magnitude-constant vector coefficient during training, while $\lambda$ in the weight decay rate of AdamW has a magnitude-varying vector coefficient during training. This provides a quantitative measure for fairly comparing the weight decay rates of AdamS and AdamW.
>
>
> Q5: I am a little confused about the statement that the effect of weight decay can be interpreted as flattening the loss landscape of $\theta$ by a factor of $(1-\eta \lambda)$ per iteration and increase the learning rate by a factor of $(1-\eta \lambda)^{-2}$ per iteration.
>
> A5: This novel interpretation can be directly derived from the equivalence of Eq 2 and Eq 3. Eq 2 describes SGD dynamics with weight decay in the original coordinates of $\theta$. If we define the rescaled coordinates of $w_{t} \equiv \theta_{t} (1-\eta \lambda)^{-t} $, we may obtain Eq 3 for $w$, where weight decay has disappeared. It means SGD dynamics with weight decay in the original coordinates is equivalent to SGD dynamics with increasing learning rate per iteration in the rescaled coordinates. Thus, weight decay can be interpreted as or replaced by iteration-wisely flattening the loss landscape and increase the learning rate. The interpretation of increasing the learning rate was also studied by Reference [2]. We will make the interpretation more clear in the next version.
>
>
> References:
>
> [1] Loshchilov, I., & Hutter, F. (2018, September). Decoupled Weight Decay Regularization. In International Conference on Learning Representations.
>
> [2] Zhang, G., Wang, C., Xu, B., & Grosse, R. (2018, September). Three Mechanisms of Weight Decay Regularization. In International Conference on Learning Representations.

---

### Official Review · AnonReviewer4 · 2020-10-24
**Interesting proposal to bridge the gap between Adam and SGD but would like to see more experiments.**

**Rating:** 5
**Confidence:** 4

**Review:**

In this paper, the authors study the effect of weight decay across different optimizers. When one uses weight decay, the learning rate which multiplies the weight decay is different from the effective learning rate (the term multiplies the gradient dependent contribution). The authors propose to adjust for this by having the effective learning rate multiplying the weight decay term, ie Delta theta= -eta_{eff} lambda theta-eta_{eff} F(gradient). For SGD, this implies that one can roughly equate weight decay and L2 regularization by rescaling the L2-parameter. For the case of Adam, the effective learning rate is anisotropic, which they fix by taking the mean of the vector 'v'. The authors show that after correcting for the effective learning rate, which they call AdamS, they can get the same performance as what one gets with SGD in CIFAR10 and CIFAR100.

I find the discussion about rho and R confusing (they are not defined precisely and using the small lr/lambda approximation seems unnecessary) and they do not really add much.  Similarly the term unstable seems a little strong given that there is nothing bad going on with training of such models. They also make comments about reinterpreting weight decay as flattening the loss and increasing the learning rate which they don't pursue further nor connect with the main point and it seems a little out of context. The main conceptual point of the paper is simply that the weight decay should have the same effective learning rate as the gradients, it would be nice if the authors could make this more clear and more central.

The paper main point then is to equate the learning rate in the weight decay coefficient by the effective learning rate and they show that this might be enough to bridge the gap between Adam and SGD. I think this is an interesting problem, but given that this is their unique point they should probably back it up with more experiments, since they only check it for CIFAR datasets (4 experiments total). If the authors could do one extra experiment on Imagenet (for example with a standard Resnet-50), that would raise my score to weak accept.

---

> ### Author Response · Authors · 2020-11-22
> **Response to Reviewer4**
>
> We appreciate the reviewer for the hard work and helpful comments.
>
> The main concerns have been duly addressed below.
>
>
>
> Q1: I find the discussion about $rho$ and $R$ confusing (they are not defined precisely and using the small $\eta$/$\lambda$ approximation seems unnecessary) and they do not really add much.
>
> A1: We will try to introduce these two concepts in a more clear way. We agree with your point about the approximations. We will remove unnecessary small $\eta$ and $\lambda$ approximations. Thanks for the helpful suggestion.
>
>
>
> Q2: They also make comments about reinterpreting weight decay as flattening the loss and increasing the learning rate which they don't pursue further nor connect with the main point and it seems a little out of context.
>
> A2: The interpretation of weight decay is becoming an important topic and is closely related to related papers. For example, the interpretation of increasing learning rate is one of core contributions in [1]. We believe our interpretation of weight decay is novel and helpful for future research in this direction.
>
>
>
> Q3: They should probably back it up with more experiments, since they only check it for CIFAR datasets (4 experiments total). If the authors could do one extra experiment on ImageNet (for example with a standard ResNet-50), that would raise my score to weak accept.
>
> A3: Thanks for the kind suggestion. We are sorry that we did not conduct experiments on ImageNet due to the limited computational resource. We will try to supply the experiments of standard ResNet50 on ImageNet as soon. We also respectfully note that Figure 16, which is mentioned in “Empirical Results” and presented in Appendix C, shows the Language Modeling experiment (LSTM on Penn TreeBank). AdamS still outperforms AdamW and Adam significantly. We will move Figure 16 into the main paper in the next version. Our paper actually studied four more models, including VGG, DenseNet, GoogLeNet, and LSTM, beyond the classical work [2].
>
> Thanks again for your helpful suggestions.
>
>
> References:
>
> [1] Zhang, G., Wang, C., Xu, B., & Grosse, R. (2018, September). Three Mechanisms of Weight Decay Regularization. In International Conference on Learning Representations.
>
> [2] Loshchilov, I., & Hutter, F. (2018, September). Decoupled Weight Decay Regularization. In International Conference on Learning Representations.

---

### Official Review · AnonReviewer3 · 2020-10-28
**Review of "Stable Weight Decay Regularization"**

**Rating:** 6
**Confidence:** 4

**Review:**

Summary:
This paper presents a novel framework that alters the weight decay update rule and aims to improve generalization when applied to 1) momentum-based optimizers and 2) adaptive optimizers. The framework includes the concept of “weight decay rate” and “total weight decay”, where the idea is to make the “weight decay rate” constant throughout training. The framework is applied to “fix” or “stabilize” weight decay when applied to SGD with momentum and Adam (which effectively amounts to changing the weight decay parameter). The result is better generalization when applied to a variety of image classification tasks, compared to the appropriate baselines.

Strengths:
The core of the idea is novel, relatively simple, and the experiments support the claim.

Weaknesses:
My main complaint is that this paper is poorly written. I don’t understand the motivation behind the reparameterization of the update rule with w instead of \theta. The concept of “weight decay rate” and “total weight decay” seems to be the most critical part of the paper, but the explanation surrounding them was messy and hard to understand. The language used is not precise at some points in the text.

At the current state of the paper, I recommend a reject because I don’t think it’s polished enough to be published.

Comments and questions:
- I think the concept of weight decay rate and total weight decay should be explained better, and earlier in the paper. Without knowing what “stable” meant, I had a hard time taking the introduction seriously, because I didn’t know what “unstable” meant exactly, and that the decoupled weight decay method was “unstable”.
- “\beta_3 = 0” should be “beta_3 = 1”. The Pytorch documentation calls 1-beta_3 dampening and its default value is zero.
- I’m not sure I understand the sentence “Although <\theta_{t-1}> is an approximated value of \theta_{t-1}, replacing \theta_{t-1} by <\theta_{t-1}> still introduces undesirable noise into learning dynamics.” What do you mean by undesirable noise? Why is replacing <\theta_{t-1}> by <\theta> justified?

===== Update =====

I have read the authors’ response, the updated paper, and the other reviews. I believe that the changes made by the authors address my concerns about the motivation and the lack of clarity; section 2 reads much better now.

It seems like the main complaints of the other reviewers are in the lack of more difficult workloads, and the lack of theory. I personally don’t find the lack of theory very important. I think the novelty comes from the simple observation, which no one to my knowledge has come to before, and the experiments support the idea empirically (which I think is what actually matters). I also find it a bit uncomfortable penalizing the authors for not running experiments on ImageNet, and I think the variety in architectures that the authors tried, compensates for this. I do agree that a more modern set of workloads (transformers, or even the same setup as AdamW) would have made the paper much stronger.

I increased my score to a 6 because I think the paper in its current form is enough to get accepted, but there are still improvements that could be done to make it much stronger.

---

> ### Author Response · Authors · 2020-11-22
> **Response to Reviewer3**
>
> We appreciate the reviewer for the hard work and helpful comments.
>
> The main concerns have been duly addressed below.
>
>
> Q1: I don’t understand the motivation behind the reparameterization of the update rule with w instead of $\theta$. The concept of “weight decay rate” and “total weight decay” seems to be the most critical part of the paper, but the explanation surrounding them was messy and hard to understand. The language used is not precise at some points in the text.
>
> A1: Our motivation is that weight decay should be coupled with the effective learning rates in practice, while the classical work [1] revealed that weight decay should be decoupled from the gradients. Considering the formal theoretical guarantee is lacked in the direction of fixing weight decay, we proposed quantitative measures for better analyzing weight decay, which is beyond existing work. As weight decay is one essentially important technique for training of deep networks, we believe our contribution is as promising as Reference [1]. We apologize for the writing problems. We will present core concepts and statements in more precise and formal language in the next version.
>
>
> Q2: The concept of weight decay rate and total weight decay should be explained better, and earlier in the paper.
>
> A2: We will try to introduce these two concepts in a more prominent way and present more formal theoretical analysis.
>
>
>
> Q3: $\beta_{3} = 0$ should be $\beta_{3} = 1$.
>
> A3: We apologize for the typo. We will correct it.
>
>
>
> Q4: I’m not sure I understand the sentence “Although $\theta_{t-1}$ is an approximated value of $\theta_{t-1}$, replacing $\theta_{t-1}$ by $\langle \theta_{t-1} \rangle$ still introduces undesirable noise into learning dynamics.”
>
> A4: The reviewer may misunderstand our point. This sentence is to explain how $L_{2}$ regularization is different from weight decay in SGD with Momentum (SGDM).
> If we use $L_{2}$ regularization in SGDM, we are actually decaying weights by reducing $\eta \lambda_{L_{2}} \beta_{3} \frac{1-\beta_{1}^{t}}{1-\beta_{1}} \langle \theta_{t-1} \rangle$, where $ \langle \theta_{t-1}  \rangle$ is the moving average of $\theta_{t-1}$. However, in weight decay, we should decay weights by reducing $\eta \lambda \theta_{t-1}$. We note that $L_{2}$ regularization uses $ \langle \theta_{t-1} \rangle$ instead of $\theta_{t-1}$.
> It means that, $L_{2}$ regularization is equivalent to weight decay with undesirable noise, which usually harms performance. While the difference between $L_{2}$ regularization and weight decay is obvious, few work pointed out that weight decay usually outperforms $L_{2}$ slightly in the presence of Momentum.
>
>
> References:
>
> [1] Loshchilov, I., & Hutter, F. (2018, September). Decoupled Weight Decay Regularization. In International Conference on Learning Representations.

---

### Official Review · AnonReviewer1 · 2020-10-28
**Review of Stable Weight Decay Regularization**

**Rating:** 5
**Confidence:** 3

**Review:**

## Summary

In this paper the authors introduce the notion of stable weight decay. The stable weight decay property can be defined in dimension 1 as follow: the effective learning rate represents an amount of time ellapsed between two iteration. The weight decay factor normalized (in log space) by the time ellasped should be constant across iterations.

From their framework, the authors also propose SGDS, a minor modification to SGD where the amount of L2 regularization is increased with momentum, in order to balance for the larger step sizes that the momentum will yield.

When applied to Adam, the authors derive AdamS, supposed to work better than the previous AdamW, which already improved how weight decay and Adam interact. AdamS has the stable weight decay property, unlike Adam or AdamW. AdamS weight decay amount is scaled by the denominator of Adam, taking the average over all dimensions to make it isotropic.

The authors test their methods on vision tasks, where Adam is known to underperform compared to SGD, and their method AdamS achieves significant gains compared to AdamW.

## Review

The reformulation introduced in equation (3) is an interesting alterntive view to look at weight decay, but I find it is not really used by the authors. The definition of the weight decay rate can be done from equation (2).
The authors introduce the notion of stable weight decay and seem to right away assume it is a desirable property. In particular, there is no theoretical justification that this is the case.

The benefit from SGDS is limited, except for hyper parameter tuning (as only the first few iterations of SGD are "unstable"), but it serves as a nice illustration of the new concept.

The part on adaptive methods is more interesting but the authors deviate significantly from their theoretical framework. Verifying the stable weight decay property is actually not optimal, because it is not isotropic. The authors trick is to average the value of the moving average of the squared gradients along all dimensions before using it to rescale the weight decay.

The experimental section of the paper focuses only on vision. It would have been interesting to see the effect of AdamS on other type of tasks.

Overall I think the idea introduced by the author is interesting, although the theory is not completely coherant. The experiments shows significant improvement but could have been on a more diverse set of tasks. Still,  I recommend acceptance.


## Remarks

- In the Introduction, talking about adaptive methods: "are a class of dominated methods to accelerate", I'm not sure what dominated means here.
- what is the point of having $\beta_3$? after equation (4), the authors say "SGD with momentum is $\beta_3=0$", that doesn't seem right, if $\beta_3=0$, then the gradient is completely ignored.

=============
Update avec rebutal and discussion with AC and reviewers.

After discussing with the others reviewers and AC, I have come to share their concerns with the overall fragility of the paper. We agreed that the methods is sound and likely to work better than AdamW, the proofs are not sufficient. In particular, the authors should strive to provide experiments on different training sets (ImageNet) with learning rate cross validation. The authors do not systematically compare across learning rates which make it hard to interpret the results as being conclusive. In fact only CIFAR-10 is evaluated with multiple learning rates.

The remark by Elya Loshchilov should also be adressed. Note that you cannot use stochastic noise as a justification, because for heavily overparameterized neural network, the amount of stochastic noise at the optimum is zero (i.e. perfect fitting of the training set). But there is another explanation: Intuitively for a fixed $\beta_2$, $v_t$ goes to zero as the current gradient goes to zero, and the ratio of the gradient by $v_t$ will converge to some constant, which prevents convergence. $v_t$ goes to zero at the same speed as the gradient but with a delay of $1 / (1 - \beta_2)$. If there is no convergence, then the gradients won't actually go to zero.
The only way to prevent $v_t$ from going to zero is to have $\beta_2 \rightarrow 1$ (i.e. the previously mentioned delay going to infinity), but in that case $\bar{v}_t$ won't go to zero neither. This is only an idea of a possible justification and I would encourage the authors to think carefully about this stability issues in the next revisions.

Finally I would encourage the authors the remove from the theoretical analysis parts that are not actually used (which I and other reviewers have noted).

---

> ### Author Response · Authors · 2020-11-22
> **Response to Reviewer1**
>
> We appreciate the reviewer for the helpful comments and kind support to our work.
> The main concerns have been duly addressed below.
>
> Q1: There is no formal theoretical justification that stable weight decay is a desirable property.
>
> A1: Thanks for the comment. We will make more formal theoretical justifications in the next version. Reference [1] revealed that weight decay should be decoupled from the gradients, and we revealed that weight decay should be coupled with the effective learning rates during training. We both successfully revealed hidden yet important properties about weight decay, although the theoretical guarantee behind the properties is an open problem. The quantitative measures which we proposed may be a contribution in this direction. We will explore the theoretical guarantee in future.
>
> Q2: The experimental section of the paper focuses only on vision. It would have been interesting to see the effect of AdamS on other type of tasks.
>
> A2: Thanks for the suggestion. We will conduct more comparative experiments on diverse tasks in future. We also respectfully note that Figure 16, which is mentioned in “Empirical Results” and presented in Appendix C, shows the Language Modeling experiment (LSTM on Penn Treebank). AdamS still outperforms AdamW and Adam significantly. We will move Figure 16 into the main paper in the next version.
>
> Q3: In the Introduction, talking about adaptive methods: "are a class of dominated methods to accelerate", I'm not sure what dominated means here.
>
> A3: We apologize for the unclear presentation. It means that adaptive gradient methods are powerful and popular. We will state it more clearly in the next version.
>
> Q4: What is the point of having $\beta_{3}$? After equation (4), the authors say "SGD with momentum is $\beta_{3}$ ", that doesn't seem right, if $\beta_{3} = 0$, then the gradient is completely ignored.
>
> A4: Thanks for pointing out the typo. It should be $\beta_{3} = 1$ in SGD (and be $\beta_{3}=1-\beta_{1}$ in Adam-style momentum). The SGD implementation in PyTorch have a hyperparameter, denoted as $1-dampening$, to tune $\beta_{3}$. We think it is great to make our theoretical analysis include this case.
>
>
>
> References:
>
> [1] Loshchilov, I., & Hutter, F. (2018, September). Decoupled Weight Decay Regularization. In International Conference on Learning Representations.

---

> ### Author Response · Authors · 2021-01-15
> **Response to Reviewer 1's new feedbacks**
>
> Thanks for the reviewer's new feedbacks.
>
> We would like to clarify your new concerns.
>
> Q5: The authors do not systematically compare across learning rates which make it hard to interpret the results as being conclusive. In fact only CIFAR-10 is evaluated with multiple learning rates.
>
> A5: We note that we did systematically compare across learning rates on various models and datasets. While we only displayed the optimal baseline (SGD) performance, we actually selected the optimal learning rates from $\{0.001, 0.01, 0.1, 1, 10\}$, seen in Appendix B.1. For Adam variants, we follow the hyperparameters recommended by original papers, as some Adam variants are claimed to outperform SGD in the recommended settings. In contrast, many related papers did not tune good baselines (SGD), but chose $\lambda=0.0001$ directly and had lower baselines than our paper. Thanks again for your suggestion. We will add more experiments to further improve our work in future.
>
> Q6: The remark by Elya Loshchilov should also be addressed. Note that you cannot use stochastic noise as a justification, because for heavily overparameterized neural network, the amount of stochastic noise at the optimum is zero (i.e. perfect fitting of the training set). But there is another explanation.
>
> A6: We respectfully note that we can use stochastic gradient noise as a justification, theoretically and empirically. The reviewer may confuse SGD with Gradient Descent. The amount of stochastic noise at the optimum is not zero and even far from zero for both heavily overparameterized neural networks and one-dimensional toy models (when the Hessian is not a zero matrix). Only for full-batch training (Gradient Descent), we have $\sqrt{\bar{v}}= 0$ at critical points. The structure of stochastic gradient noise is a very interesting topic. Related papers [2,3,4], which focused on investigating the structure of stochastic gradient noise, theoretically and empirically revealed that the covariance of stochastic gradient noise is approximately proportional to the Hessian and inverse to the batch size at/near minima. Particularly, the variance of stochastic gradient is directly observed to be much larger than the expectation of stochastic gradient at/near minima. We are sorry that we did not expect the misunderstanding about stochastic gradient noise is not rare. We will add discussion about this point into the paper in future.
>
>
> References:
>
> [2] Jastrzębski, S., Kenton, Z., Arpit, D., Ballas, N., Fischer, A., Bengio, Y., & Storkey, A. (2017). Three factors influencing minima in sgd. arXiv preprint arXiv:1711.04623.
>
> [3] Zhu, Z., Wu, J., Yu, B., Wu, L., & Ma, J. (2019, June). The Anisotropic Noise in Stochastic Gradient Descent: Its Behavior of Escaping from Sharp Minima and Regularization Effects. In ICML (pp. 7654-7663).
>
> [4] Daneshmand, H., Kohler, J., Lucchi, A., & Hofmann, T. (2018, July). Escaping Saddles with Stochastic Gradients. In International Conference on Machine Learning (pp. 1155-1164).

---

### Public Comment · ~Ilya_Loshchilov1 · 2020-11-16
**Comment**

In AdamS, the weight decay factor of AdamW is divided by the average moving gradient amplitude of all variables (square root of mean v_hat). That average moving gradient amplitude should tend to 0 closer to the optimum, thus the weight decay contribution will tend to infinity due to the division. It might be problematic. Perhaps, one could present a toy problem (e.g., one-dimensional for the sake of simplicity) where AdamS does not converge (one might need an epsilon parameter to be added to resolve it).

The curve of square root of mean v^hat_t over time could probably demonstrate the additional scheduler for weight decay which the proposed modification represents. Some schedulers for weight decay were studied by Smith.

The paper does not use cosine annealing (also see [1]) which was used for AdamW and many recent works with CIFAR datasets. It might be more important than it seems, see below.
Please consult Figure 1 in https://arxiv.org/pdf/1711.05101.pdf.
Top row middle column subfigure shows the results for a step-drop learning rate schedule similar to the one used in your paper.
Top row right column subfigure shows the results for cosine annealing as used in the original AdamW.
The difference between AdamS and AdamW shown in Figure 8 of the paper is comparable to the one in the two subfigures mentioned above.
Thus, the difference shown between AdamW and AdamS might potentially be explained by the lack of cosine annealing (and its smoothness which in AdamS might be partially compensated by the introduced gradient norm based multiplier for weight decay).

It would be great to use more powerful networks. The ones used for CIFAR-10 provide larger error rates (4.5%) than the ones used in the original AdamW paper (3.2%) that appeared 3 years ago.

[1] A Second look at Exponential and Cosine Step Sizes: Simplicity, Convergence, and Performance. by Li, Zhuang and Orabona.
https://arxiv.org/pdf/2002.05273.pdf.

---

> ### Author Response · Authors · 2020-11-22
> **Response to "Comment" of Dr. Ilya Loshchilov**
>
> We gratefully appreciate your helpful comments.
>
> Your paper “Decoupled Weight Decay Regularization” inspired us to further analyze weight decay.
> It is our pleasure to clarify your concerns.
>
> Q1: In AdamS, the weight decay factor of AdamW is divided by the average moving gradient amplitude of all variables (square root of mean v_hat). That average moving gradient amplitude should tend to 0 closer to the optimum, thus the weight decay contribution will tend to infinity due to the division.
>
> A1: We respectfully argue that, due to stochastic gradients, $\sqrt{\bar{v}}$ is not expected to get close to 0 near minima. Note that $\sqrt{\bar{v}}$ actually depends on the gradient variance instead of the gradient mean. While the gradient mean is indeed expected to be 0 near minima, the gradient variance depends on the Hessian near minima. It is also well known that the gradient variance dominates the gradient mean near minima. Empirically, we observed that $\sqrt{\bar{v}}$ is around $10^{-3}$ in the final phase of training on CIFAR-10 and CIFAR-100. Even in a toy one-dimensional case, the $\sqrt{\bar{v}}$ will not converge to 0, as long as the Hessian near minima is not a zero matrix. Thus, it is optional but not necessary to have such an $\epsilon$ in AdamS.
>
> Q2: The curve of square root of mean $\bar{v}_{t}$ over time could probably demonstrate the additional scheduler for weight decay which the proposed modification represents. Some schedulers for weight decay were studied by Smith.
>
> A2: Thanks for the comment. Our paper suggests that weight decay should be coupled with the effective learning rate, which is indeed a special scheduler. Our method has two obvious advantages beyond artificial schedulers. First, SWD does not require any extra hyperparameter like conventional schedulers. Second, the weight decay rate is a reasonable quantitative measure from a perspective of learning dynamics. We think it will be interesting to study existing weight decay schedulers in terms of the weight decay rates we proposed.
>
> Q3: The paper does not use cosine annealing (also see [1]) which was used for AdamW and many recent works with CIFAR datasets. The difference shown between AdamW and AdamS might potentially be explained by the lack of cosine annealing (and its smoothness which in AdamS might be partially compensated by the introduced gradient norm based multiplier for weight decay).
>
>
> A3: Although the dependence of various optimizers on various learning rate schedulers is beyond the scope of our main contribution, we agree that it is always great to have more empirical evidence. We are very willing to supply the experiments which use cosine annealing and warm restarts schedulers in the next version. Our recent results of ResNet18 and VGG16 both suggest that AdamS still yields significantly better performance than AdamW and Adam with cosine annealing and warm restarts. We will present the complete experimental results in the next version once the supplementary experiments are finished.
>
>
> Q4: It would be great to use more powerful networks. The ones used for CIFAR-10 provide larger error rates (4.5%) than the ones used in the original AdamW paper (3.2%) that appeared 3 years ago.
>
> A4: Thanks again for the helpful suggestion. We will supply the experiment on larger models in future. We think that the diversity and the accuracy are both important. While ResNet18 has a lower baseline performance than 26 2x64d ResNet, our baseline performance, including ResNet, VGG, DenseNet, and GoogLeNet, is higher than the popular baseline performance of the corresponding models.

---

### Author Response · Authors · 2020-11-22
**General Response**

We appreciate all reviewers for the hard work and helpful comments.
We would like to address all reviewers’ concerns in the corresponding responses.

We also hope to emphasize our main contribution.
Our paper proposed that weight decay should be coupled with the effective learning rates in practice, while the classical work [1] revealed that weight decay should be decoupled from the gradients. We also proposed new quantitative measures for better analyzing weight decay, which is another contribution beyond previous work. As weight decay is one essentially important technique for training of deep networks, we believe our paper is comparable with the classical work [1] in the sense of significance and novelty.


We have revised our manuscript according to the comments. The change we made mainly includes:

-	We introduced two key concepts in a more prominent way.
-	We presented core statements 1-4 more formally. We proposed several propositions in the main paper and added the corresponding proofs in the appendix.
-	We particularly presented more theoretical analysis of AdamS.
-	We moved the language modeling experiment into the main paper.
-	We added the experiments of AdamS, AdamW, and Adam with cosine annealing schedulers and warm restarts in the appendix.


Reference:

[1] Loshchilov, I., & Hutter, F. (2018, September). Decoupled Weight Decay Regularization. In International Conference on Learning Representations.

---

### Decision · Program_Chairs · 2021-01-07
**Final Decision**

**Decision:**

Reject

**Comment:**

The paper proposes a novel way to have weight decay-like update rule. Empirically, the authors claim that it improves generalization when applied to momentum-based optimizers and optimizers with coordinate-wise learning rates.

This paper has been thoroughly discussed, both in public and private mode.
The strength of this paper lies in the possible gain in generalization performance due the proposed change.
The weaknesses are:
- the very confusing and not scientific motivation of the proposed change
- the experiments are not fully convincing

More in details, we all found the discussion on "stable" and "unstable" weight decay extremely confusing. The claim of the paper is that "stable" weight decay should be preferred over "unstable" one. However, to validate a scientific claim it is necessary to carry out an empirical or theoretical evaluation. The theoretical one is simply missing: a number of proposition and corollaries are stated with some simple mathematical facts completely disconnected from the optimization or generalization issues. As it is, removing these arguments would actually make the paper better.
On the empirical side, there is no experiment that supports the simple claim that "the unstable weight decay problem may undesirably damage performance". Instead, what we see are experiments in which the modified update rule seem to perform better, but they don't actually show that "stability" or "instability" are the specific issues at play here. Indeed, any other explanation is equally valid and the experiments do not support any specific one, but rather they can only support the claim that the proposed algorithm might be better than some other optimization algorithms. The *specific reason* why this is happening is not clear.

Turning to the empirical evaluation, the discussion elicited the fact that, a part for CIFAR10, the experiments are carried out without tuning of the learning rates. Hence, it is difficult for us to even validate the claim of superiority of the method. I don't subscribe to the idea that a deep learning paper requires experiments on ImageNet to be valid. Yet, given that there is no supporting theory in this paper, the empirical evaluation should be solid and thorough.

For the above reasons, the paper cannot be published at ICLR.